# Outcome Comparison of Drug-Resistant Trigeminal Neuralgia Surgical Treatments—An Umbrella Review of Meta-Analyses and Systematic Reviews

**DOI:** 10.3390/brainsci13040530

**Published:** 2023-03-23

**Authors:** Alessandro Rapisarda, Marco Battistelli, Alessandro Izzo, Manuela D’Ercole, Quintino Giorgio D’Alessandris, Filippo Maria Polli, Samuele Santi, Renata Martinelli, Nicola Montano

**Affiliations:** 1Department of Neurosurgery, Fondazione Policlinico Universitario Agostino Gemelli IRCSS, 00168 Rome, Italy; alessandro.rapisarda@guest.policlinicogemelli.it (A.R.); alessandro.izzo@policlinicogemelli.it (A.I.); manuela.dercole@policlinicogemelli.it (M.D.); quintinogiorgio.dalessandris@policlinicogemelli.it (Q.G.D.); filippomaria.polli@policlinicogemelli.it (F.M.P.); 2Department of Neuroscience, Neurosurgery Section, Università Cattolica del Sacro Cuore, 00168 Rome, Italy; marco.battistelli23494@gmail.com (M.B.); santisamuele@yahoo.it (S.S.); renata.martinelli01@icatt.it (R.M.)

**Keywords:** trigeminal neuralgia, microvascular decompression, percutaneous procedures, radiosurgery, umbrella review

## Abstract

Medical treatment for trigeminal neuralgia (TN) is not always a feasible option due to a lack of full response or adverse effects. Open surgery or percutaneous procedures are advocated in these cases. Several articles have compared the results among different techniques. Nevertheless, the findings of these studies are heterogeneous. Umbrella reviews are studies sitting at the peak of the evidence pyramid. With this umbrella review, we provided a systematic review of the outcomes of the surgical procedures used for TN treatment. Only systematic reviews and meta-analyses were included following the PRISMA guidelines. Ten articles were enrolled for qualitative and quantitative assessment. Level of evidence was quantified using a specific tool (AMSTAR-2). Results were heterogenous in terms of outcome and measurements. Microvascular decompression (MVD) appeared to be the most effective procedure both in the short-term (pain relief in 85–96.6% of cases) and long-term follow-up (pain relief in 64–79% of cases), although showed the highest rate of complications. The results of percutaneous techniques were similar but radiosurgery showed the highest variation in term of pain relief and a higher rate of delayed responses. The use of the AMSTAR-2 tool to quantify the evidence level scored three studies as critically low and seven studies as low-level, revealing a lack of good quality studies on this topic. Our umbrella review evidenced the need of well-designed comparative studies and the utilization of validated scales in order to provide more homogenous data for pooled-analyses and meta-analyses in the field of TN surgical treatment.

## 1. Introduction

Trigeminal neuralgia (TN) is a form of neuropathic facial pain that greatly impacts the quality of life of the affected patients [1]. It is usually described as a sudden pain referred as a stab or electric shock in the area of distribution of one or more trigeminal branches. Typical TN usually presents as relapse-remitting pain with periods of total pain relief (now referred as TN with pure paroxysmal pain), whereas atypical TN comes with continuous or sub-continuous pain (now referred as TN with continuous pain) [2]. From an etiological point of view, TN can also be distinguished in primary TN or secondary TN. In the latter case, multiple sclerosis, cerebello-pontine angle tumors, and vascular malformations are responsabile for trigeminal nerve compression or damage. Primary TN can be further divided in a classical form when a vessel (more often an artery as the superior cerebellar artery or the antero-inferior cerebellar artery) is found to present a neurovascular conflict (NVC) with the trigeminal nerve, and in an idiopathic form when the evidence of a NVC or any other suitable cause is lacking. A summary of the pathophysiological theories and molecular mechanisms involved in TN is provided in Figure 1. From a molecular point of view, different molecules have been advocated to be involved in TN pathogenesis. Recently, we investigated the role of different inflammatory and neurodegenerative factors and found that neuron specific enolase (NSE) is involved in TN pathogenesis [1]. NSE is a glycolytic enzyme that is expressed in the cytosol of neurons and neuroendocrine cells. Elevated NSE can promote extra-cellular matrix degradation, inflammatory glial cell proliferation, and actin remodeling, thereby affecting the migration of activated macrophages and microglia to the injury site and promoting neuronal cell death. Different types of insults may damage trigeminal nerve, leading to the degeneration and axonal dystrophy of Schwann’s cells. According to our results, NSE could be upregulated in TN, thus transiting on the cell surface of neuronal cells where it contributes to trigger nerve damage and inflammation. In fact, we found elevated NSE in the serum of TN affected patients [1]. Furthermore NSE concentration in CSF appeared strictly related to the severity of NVC, being highly expressed in the case of nerve atrophy [3]. While being effectively treated with medications in most cases, TN may become progressively drug-resistant or patients can experience severe drawbacks from the drug-intake. These cases are considered for surgical treatment through open surgery (microvascular decompression, MVD) or palliative percutaneous destructive procedures (percutaneous balloon compression, PBC; percutaneous glycerol injection, PGI; radiofrequency rhizotomy, RF; gamma-knife surgery, GKS; stereotactic radiosurgery, SRS). Other types of procedures include the peripheral blockade of trigeminal nerve branches by neurectomy, alcohol injections, radiofrequency, or cryolesions, but their effectiveness is not supported by clinical trials [4] (Figure 2) Some meta-analyses and systematic reviews have compared the different outcomes and complications of these techniques, however, the number of studies included is scarce and the strength of evidence of the results is debatable.

An umbrella review is a study that allows one to summarize the existing literature evidence on a certain topic. The most typical feature of this type of evidence synthesis is that only studies with the highest level of evidence (systematic reviews, meta-analyses) can be considered for inclusion [5].

With this umbrella review, we aimed at providing an updated evaluation of the evidence reported in the literature and compared the results of the different surgical procedures.

## 2. Materials and Methods

### 2.1. Enrollment

Systematic reviews and systematic reviews with meta-analysis comparing different interventional techniques for the treatment of TN were collected through multiple database screening (PubMed, Scopus, Cochrane). The terms “surgery”, “MVD”, “microvascular decompression”, “percutaneous”, “PBC”, “balloon compression”, “RFR”, “RF” “radiofrequency rhizotomy”, “thermoablation, “PGI”, “percutaneous glycerol injection”, “alcoholization”, “cryotherapy”, “GKS”, “gamma-knife surgery”, “stereotactic radiosurgery”, “SRS”, ”trigeminal neuralgia”, and “tic douloureux” were searched on the above-mentioned databases in every possible combination, without year limitation (last search lunched on January 2023). Two independently working authors (A.R.; M.B.) went through the screening by title, identifying 101 fitting articles. A senior author (N.M.) provided supervision of the screening process by abstract, thus enrolling 12 articles eligible for full-text evaluation. Of these, two were excluded because the study design lacked comparative results. The study flow diagram is depicted in Figure 3 according to the PRISMA guidelines [6].

Of the 10 enrolled studies, three were systematic reviews [7,8,9] and seven were meta-analyses [10,11,12,13,14,15,16], as reported in Table 1.

### 2.2. Level of Evidence Assessment

The eligible studies were thoroughly assessed with the AMSTAR-2 tool (updated version of the AMSTAR, A Measurement Tool to Assess Systematic Reviews) [17] to quantify the strength of the evidence. Hence, the articles were categorized into one of the four levels of evidence: high, moderate, low, and critically low (Table 2).

All numeric data were reported as the median. A *p*-value (*p*) under 0.005 was considered as statistically significant.

## 3. Results

The included reviews varied with respect to the search strategies (database searched), the type of studies included (randomized clinical trial, RCTs; non-randomized clinical trial NRCTs), and review styles. Most of the included studies were observational ones, with few reported RCTs [7,9]. The included studies compared the different techniques as follows: MVD [7,8,10,11,12,13,15,16], GKS [7,8,10,12], rhizotomy techniques (PBC, RFR, PGI) [7,9,10,11,13,14], SRS [7,9,16], cryotherapy [7], and endoscope-assisted MVD [10,15]. The systematic reviews and meta-analyses differed with regard to the number of included studies, from the 43 studies included by Mendelson et al. [8] to the five included by Sharma et al. [12]; thereafter, the number of patients differed widely among the studies. Moreover, the follow-up (FU) range significantly varied from 12 to 204 months for Yan et al. [11] to 5–30 months for Texakalidis et al. [14]. Another source of heterogeneity comes from the evaluation of the outcomes, especially regarding pain-free measurements, since not all the studies used a standardized method as the BNI (Barrow Neurological Institute, Phoenix, AZ, USA) scale to measure that outcome. Table 1 summarizes the details of the included studies.

### 3.1. Microvascular Decompression

Concerning pain relief, all but one of the included studies [9] assessed pain relief immediately after the procedure (acute pain relief, APR). MVD was reported as the most effective treatment in the short-term period, with similar results among the different meta-analyses. Diana et al. [7], in the studies included, reported an APR rate of 85–96.6%, with similar results reported in the studies of Mendelson et al. [8] and Sharma et al. (96%) [12]. A subgroup analyses in patients without multiple sclerosis (MS) reported a statistically significant (*p* < 0.05) higher rate of APR in MVD (86.7–97%) when compared with PBC (72–96%) with an OR of 0.54 (CI 95%: 0.34–0.84) [13].

Chen et al. [10] reported a 9.6% recurrence rate, while Mendelson et al. [8] reported a 14.93% (CI 95%: 0.09–0.21) recurrence rate at last FU (23.4–300 months). Lu et al. [16] stated a significantly higher chance of pain relapse in SRS than in MVD in a long-term period (6–36 months; SRS vs. MVD OR 0.29; CI 95% 0.19–0.46). Pain-free rate at ten years was assessed among a range of 64–74% by Diana et al. [7], similar to the outcomes described by Mendelson et al. [8] at the last available FU (79.37%, CI 95%: 0.75–0.83; FU 23.4–300 months). Sharma et al. [12] reported a 3–5 year MVD success rate of 72%. In the subgroup without MS, Nascimento et al. reported a trend toward better outcome in long-term follow-up (FU; 6–168 months) for MVD (58–91.2%) with respect to PBC (70–85.8%), with an OR of 0.56 (CI 95%: 0.27–1.13). Recurrence rate analyses showed that patients who underwent the MVD procedure had lower recurrence than those who underwent PBC, with an OR of 0.61 (CI 95%: 0.29–1.26) [13].

Postoperative complications were reported far more frequently for MVD than other less invasive procedures. More specifically, Lu et al. [16] reported higher postoperative complications in MVD compared with SRS (OR 0.05; CI 95% 0.01–0.18). In this study, the most common reported complication was CSF leak, with an incidence of 12 out 577 MVD procedures while the other reported complications were wound infection, pseudomeningocele, hematoma, facial paralysis, hearing loss, pneumonia, deep venous thrombosis, diplopia, transverse sinus injury, and cerebral infarction. Diana et al. [7] reported a 4–28% rate of facial numbness and a 14.7% incidence of herpes simplex infection. However, Lu et al. [16] showed a statistically significant more frequent postoperative facial numbness in SRS (OR 2.04; CI 95% 1.22–3.41).

### 3.2. Endoscope-Assisted MVD

Two of the ten considered studies reported a comparison between microscopic MVD and endoscopic MVD [10,15]. Zagzoog et al. [15] reported an APR in 88% (CI 95% 83–93%) of cases for endoscopic MVD vs. 81% (CI 95% 74–86%) of cases for microscopic MVD. Recurrence rate appeared to be favorable in endoscopic MVD with respect to microscopic MVD: 9% (CI 95% 5–14%) vs. 14% (CI 95% 8–21%). Chen et al. [10] reported a 9.6% recurrence rate for microscopic MVD, while reporting a 2.3% recurrence rate for endoscopic MVD. Complication rates appeared to also be less frequent in endoscopic MVD, particularly with respect to facial paresis (microscopic MVD 9%, CI 95% 4–16%; endoscopic MVD 3%, CI 95% 0–8%), and hearing loss (microscopic MVD 4%, CI 95% 2–6%; endoscopic MVD 1%, CI 95% 0–3%), while no significant differences were appreciated for the CSF-leakage rate (microscopic MVD 3%, CI 95% 1–6%; endoscopic MVD 3%, CI 95% 2–4%) [15].

### 3.3. Stereotactic Radiosurgery and Gamma-Knife Surgery

Three systematic reviews and meta-analyses focused on the SRS effects [7,9,16]. The pain relief rate was assessed by Lopez et al., with a six months pain relief rate ranging from 66% to 68% and a three year pain relief of 55–56% [7]. Lu et al. stated that SRS never showed as high as the MVD pain relief odds over time, neither in the short-term (OR 0.12; CI 95%: 0.08–0.18) nor in the long-term (OR 0.29; CI 95% 0.19–0.46). Even in pain-free survival, SRS proved to be less effective than MVD (OR 2.28; CI 95% 1.25–4.29). Even if SRS showed less frequent complications compared to MVD (OR 0.05; CI 95% 0.01–0.18), facial numbness and dysesthesia were statistically more frequent in SRS according to Lu et al. (OR 2.04; CI 95% 1.22–3.41) [16].

Four meta-analyses took GKS into consideration [7,8,10,12]. Diana et al. [7] showed a huge variability in the short-term pain relief among the included studies, since pain relief ranged between 23% and 96.2% [18,19,20,21,22]. Mendelson et al. [8] and Sharma et al. [12] showed a statistically significant better short-term pain control in MVD compared to GKS, with a pain relief rate of 61.45% (CI 95%: 0.38–0.82) [8] and 72% [12], respectively. The GKS recurrence rate ranged from 19.38% (CI 95%: 0.13–0.27) in Mendelson et al. [8] to 0.9–51.9% in Diana et al. [7]. The long-term success rate appeared to be similar between these two meta-analyses with Sharma et al. reporting a mean 3 to 5 year success rate of 46% [12], while Mendelson et al. reported a 41.62% (CI 95%: 0.35–0.49) success rate at last follow-up (10–90 months) [8]. Complications were recorded among studies, the most common of which appeared to be facial numbness (0–36.5% [7], 24% [12]). Other common complications were severe dry eye, loss of corneal reflex, and dysesthetic pain [7,12].

### 3.4. Percutaneous Balloon Compression

PBC was considered in five of the included meta-analyses [7,9,10,11,14]. Diana et al. and Lopez et al. showed a similar short-term pain relief rate (87% and 91%, respectively), even if only three substudies in Diana et al. [19,23,24] and one substudy in Lopez et al. [25] took PBC into consideration [7,9]. Taxakalidis et al. did not show a significant difference in short-term pain relief comparing PBC with PGI (OR 2.31; CI (95%) 0.94–5.70) or PBC and RF (OR 0.63; CI (95%) 0.23–1.74) nor in pain recurrence between PBC and PGI (OR 0.52; CI (95%) 0.23–1.21) (FU 6–28.5 months) or PBC and RF (OR 0.72; CI (95%) 0.35–1.47) (FU 5–29 months) [14]. The pain recurrence rate ranged from as high as 45.2% in Diana et al. [7] to as low as 12.3% in Chen et al. [10]. Three year pain relief was 69% in a single study [25] considered in the meta-analysis by Lopez et al. [9]. The complication rate was 16.1% with a transient masticatory weakness reported as high as 100% of cases by Lopez et al. [9]. Other reported complications were herpes simplex infection, facial numbness, dysesthesia, IV and VI cranial nerve palsy, and meningitis. Texakalidis et al. reported a significantly higher risk of mastication weakness (OR 9.29; CI (95%) 2.71–31.86) and diplopia due to CN IV or VI palsy (OR 6.31; CI (95%) 1.70–23.33) when PBC was compared to PGI, while no significant differences were noted in comparison with RF [14].

### 3.5. Radiofrequency Rizotomy

The same five meta-analyses took RF into consideration [7,9,10,11,14]. Three studies included by Lopez et al. reported the six month pain free rate as high as 74–94% [26,27,28]. On the other hand, deterioration of a positive effect of RF was appreciated between one year FU (70–90%) and two years FU (62–65%); final pain relief assessment at 5 years showed a 51–56% success rate [9]. Yan et al. and Texakalidis et al. showed higher odds of immediate pain relief compared to PGI (OR 2.65; CI (95%) 1.29–5.44) and PBC (OR 2.65; CI (95%) 1.29–5.44), respectively [11,14]. Sensitivity analyses after excluding Meglio et al. and Noorani et al. [29,30] showed higher odds of immediate pain relief (OR 2.53; CI (95%) 1.04–6.19) and a reduced risk of recurrence compared to PBC (OR 0.71; CI (95%) 0.50–1.00) [11]. However, the same study reported an increased risk of pain recurrence in comparison to MVD (OR 3.80; CI (95%) 2.00–7.20). Diana et al. reported a facial numbness rate after the RF procedure as high as 50% [7] and Lopez et al. reported a complication frequency of 29.2% [9]. Both Yan et al. and Texakalidis et al. reported higher odds of facial anesthesia compared to PGI (OR 3.01; CI (95%) 1.11–8.13) and MVD (OR 4.62; CI (95%) 2.15–9.93), but not when compared to PBC (OR 3.40; CI (95%) 0.01–1166.52), and a lower risk of herpes eruption compared to PGI (OR 0.30; CI (95%) 0.17–0.56) [11,14].

### 3.6. Percutaneous Glycerol Injection

A study included by Diana et al. [7] and two studies included by Lopez et al. [9] showed a similar rate of immediate pain relief (85% [24] and 78–88% [31,32], respectively), with a decline at three years to a 53–54% pain relief rate [7,33]. As previously mentioned, PGI had a lower odds of immediate pain relief compared to RF [13,16], but not compared to PBC [14], and a comparable odd of pain recurrence with respect to both RF and PBC [11,14]. The overall pain recurrence rate was 12.4% according to Chen et al. [10] and the overall complication rate was 24.8% according to Lopez et al. [33]. The most common complications that appeared were masticatory weakness, facial dysesthesia, anesthesia dolorosa, and corneal numbness. However, GR showed significantly lower odds of postoperative anesthesia compared to RF (RF vs. PGI OR 3.01; CI (95%) 1.11–8.13) [11], (RFR vs. PGI OR 4.73; CI (95%) 2.25–9.96) [14], and of masticatory weakness (PBC vs. PGI OR 9.29; CI (95%) 2.71–31.86) and diplopia compared to PBC (PBC vs. PGI OR 6.31; CI (95%) 1.70–23.33) [14].

### 3.7. Study Quality Assessment

The AMSTAR 2 rating score was used to assess the quality of the included study [17]. None of the included studies reached a high or moderate quality score, and three of them scored critically low [8,11,15]; all of the others gained a low quality score [7,9,10,11,12,13,14,16].

## 4. Discussion

In 1934, Walter Dandy postulated the pathogenesis of TN as related to vascular compression [34]. After the introduction of vascular decompression by Gardner [35], and later, of the microsurgical approach to NVC as described by Jannetta [36] and Barker [37], MVD became the gold standard for the treatment of TN. However, MVD is indicated in patients able to sustain an open neurosurgical procedure and is far more suited in primary classical TN when evidence of a NVC occurs. Other forms of TN or patients not suitable for open surgery are considered for percutaneous approaches that have been compared in terms of the outcomes and drawbacks in recent literature. The present umbrella review aimed at providing an overview on the surgical treatments for drug-resistant TN in terms of pain-relief, pain relapse, and complication rate. A total of 10 meta-analyses and systematic reviews were included. A critical point emerging from our umbrella review was that the assessment of the quality of the included studies revealed a severe lack of evidence, as none of them reached a moderate or a high quality in AMSTAR2 grades. However looking to the overall results of the included studies, MVD appeared to be the most effective treatment in short- and long-term FU, with the lowest recurrence rate. Nonetheless, it showed the highest rate of complications, although the routine use of neuronavigation seems to reduce the complication rate and extend the utility of MVD, even in elderly patients [38]. Endoscopic MVD has emerged as a possible alternative to microsurgical MVD because of a reduced morbidity rate, even if longer and specific training may be required. More data clearly emerged from our umbrella review: among the percutaneous treatments, GKS and SRS showed the highest variation in terms of pain relief among the different studies and the highest amount of delayed responses. Improvement score, as assessed by BNI, appeared to be linearly correlated with an increased dose from 80 Gy to 90 Gy and was greater for patients who were treated with two shots compared to patients treated with one shot [39]. Moreover, we found a wide variation among studies with regard to the optimal radiation dose, location of delivery, number of isocenters, and the length of the involved trigeminal nerve for radiation [8]. Thus, the standardization of these techniques is, in our opinion, urgently required. Furthermore, our umbrella review confirmed that the other percutaneous techniques were equivalent in terms of the short- and long-term results, recurrence rate, and complications. With these techniques, an excellent short-term outcome could be obtained (comparable to the one obtained by MVD), but there was a higher recurrence rate in the long-term FU compared to the MVD.

The main problem in handling these results is that potential selection and recall biases were present, since most of the studies reporting TN surgical treatment were retrospective-observational ones in which the surgical procedure was chosen based upon the surgeon’s experience or the patient’s preference. Furthermore, most of the articles on TN surgical treatment have not reported whether the patients treated with each different procedure were “naïve” (never submitted to any surgical procedure for TN, except medications) or had already undergone a surgical procedure. Thus the conclusions of the majority of these articles are that TN treatment requires a patient-tailored approach and the authors themselves claim a lack of RCTs in order to support one treatment over another [7,11,13,14]. Regarding the outcome assessment, a uniform method to quantify pain should be advocated. Moreover, many studies did not specify whether patients were under medications at the time of evaluation, introducing a limitation in subgroup analyses. For example, Mendelson et al. addressed as ‘’good’’ the patients whose symptoms improved without specifying medication usage [8]. The BNI pain scale is a simple method for stratifying patients into five objective categories that consider medication use at the time of evaluation. Introducing a standardized pain evaluation scale would allow one to compare these techniques in terms of pain outcome. Another problem in evaluating the outcome after a surgical procedure for TN is the lack of consideration of the psychological aspects. A systematic review by Nova et al. [40] stated that there was limited evidence of the psychometric performance of patient-reported outcomes for TN. Sandhu et al. [41] reported on the minimum clinically important difference (MCID) interfering with activities of daily life (ADL) in facial pain using the Brief Pain Inventory Score (BPI). They concluded that pain’s interference with ADL could be more important for patients (when designing or evaluating interventions in the field of TN) instead of simply assessing the presence or not of pain.

## 5. Conclusions

Although the quality of evidence is homogenously scarce, in our umbrella review considering only systematic reviews and meta-analyses, MVD appeared to be the most effective surgical treatment for TN in both the short- and long-term FU. Nevertheless, the fact that the included studies scored low or critically low in the methodology assessment does not permit to establish a strong evidence in support of one surgical technique over another. Methodological uniformity should be warranted in future studies to achieve comparable data. RCTs comparing the different surgical techniques are advocated in order to perform high quality meta-analyses to increase the level of evidence in this field.

## Figures and Tables

**Figure 1 brainsci-13-00530-f001:**
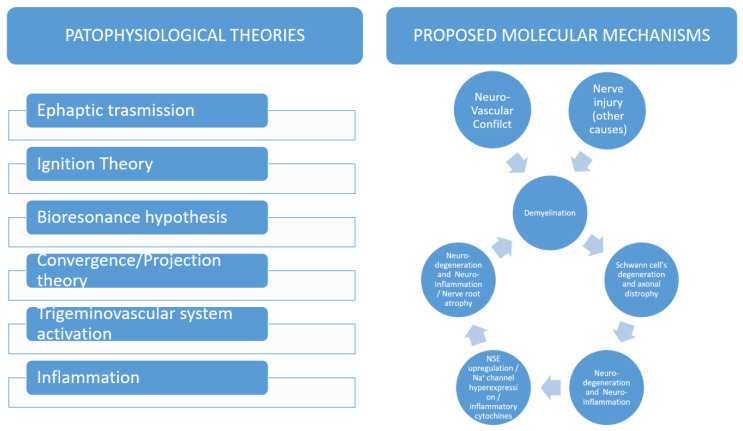
Summary of the pathophysiological theories and molecular mechanisms involved in TN pathogenesis.

**Figure 2 brainsci-13-00530-f002:**
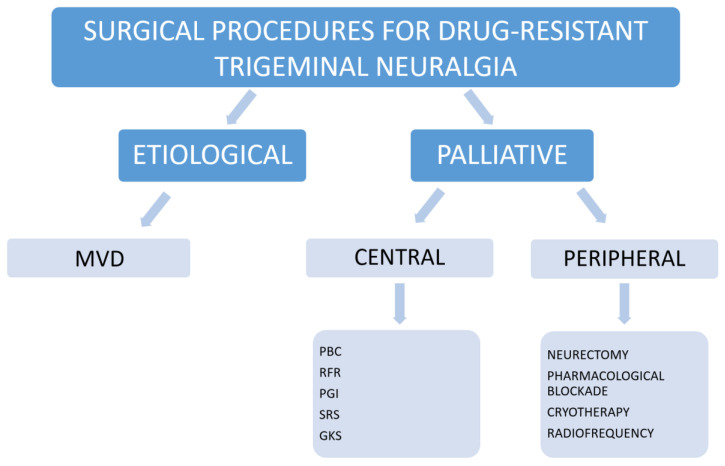
Summary of the available surgical procedures for the treatment of drug-resistant TN.

**Figure 3 brainsci-13-00530-f003:**
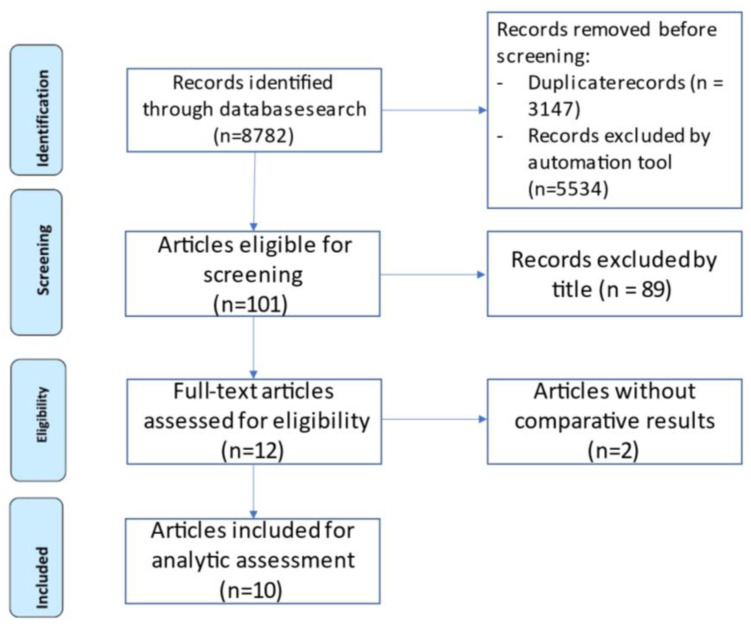
Systematic review flow diagram. The PRISMA flow diagram for the systematic review detailing the database searches and the number of articles screened for the title, abstract, and full-text.

**Table 1 brainsci-13-00530-t001:** Summary of the 10 studies included in this umbrella review.

Author (Year)	Type of Study	Number of included Studies	Compared Techniques	Conclusion
Lopez BC et al. (2004) [9]	Systematic review	11	PBC RF PGI SRS	RF: superior to PGI and SRS in terms of early and late rates of complete pain relief; highest rate of complications. PGI: superior to SRS in terms of early pain relief; the least effective technique after 24 months.
Zagzoog N et al. (2018) [15]	Meta-analysis	23	Microscopic MVD Endoscopic MVD	Endoscopic MVD pain relief and recurrence rate is comparable to microscopic MVD. Endoscopic MVD shows statistically significant lower overall rates of complications.
Lu VM et al. (2018) [16]	Meta-analysis	8	MVD SRS	MVD: greater short- and long-term pain freedom, lower incidence of facial numbness, dysesthesia and pain recurrence; higher postoperative complication rate.
Sharma R et al. (2018) [12]	Meta-analysis	5	MVD GKS	MVD superior than GKS at all durations of follow-up to 5 years in terms of pain relief. Facial numbness and dysesthetic pain rate higher in GKS procedure.
Mendelson ZS et al. (2018) [8]	Systematic review	43	MVD GKS	MVD is more effective than GKS in terms of initial pain-free and long-term pain free; also, it shows a trend toward a lower recurrence rate.
Texakalidis P et al. (2019) [14]	Meta-analysis	14	PGI RF PBC	RF: higher rate of immediate pain relief as compared to PGI, with increased risk of anesthesia and decreased risk of herpes eruption. PBC: high risk of mastication weakness and CN IV and VI palsy compared to PGI. PBC and RF do not demonstrate significant differences in terms of efficacy and safety outcomes.
Diana C et al. (2021) [7]	Systematic review	10	MVD GKS PBC PGI RF SRS Cryotherapy RT	MVD: highest success rate, low recurrence and long-term pain relief, highest complication rates. PBC: comparable initial pain relief but only for a short duration. GKS: long interval for initial pain relief, high recurrence rate, few complication rate. Lack of good studies to support one treatment over the others.
Chen JN et al. (2021) [10]	Meta-analysis	42	MVD PSR GKS PBC RF Improved MVD MVD + PSR Endoscopic MVD	MVD had the lowest recurrence rate.
Yan C et al. (2021) [11]	Meta-analysis	18	RF PGI MVD PBC	RF: safe and superior or equivalent to PBC or PGI with concern to pain relief and recurrence rate whilst inferior to MVD.
Nascimento RFV et al. (2023) [13]	Meta-analysis	11	MVD PBC	MVD appears superior to PBC in terms of short and long pain relief, recurrence of pain, and total complications.

**Table 2 brainsci-13-00530-t002:** Potential biases and level of evidence (AMSTAR-2) of the 10 studies included in this umbrella review.

Author (Year)	Potential Biases	AMSTAR-2
Lopez BC et al. (2004) [9]	Heterogeneous nature of data. Lack of standardized outcome measure (risk of subjective bias) and reporting methods. Low evidence level of the included studies.	Low
Zagzoog N et al. (2018) [15]	High attrition rate. Heterogeneity between the studies and poor availability of comparable data. Lack of standardized outcome measure. Heterogeneity in population numbers and group sizes.	Critically low
Lu VM et al. (2018) [16]	Low evidence level of the included studies. Short follow-up of some included studies. Lack of standardized outcome measure: risk of subjective bias.	Low
Sharma R et al. (2018) [12]	Lack of uniformity regarding the duration of sustained pain relief. Lack of information regarding attrition during follow-up. Lack of standardized outcome measure: risk of subjective bias. Possible ‘file drawer effect’ in some included studies.	Low
Mendelson ZS et al. (2018) [8]	Quality of available data. Possible institutional, geographic, selection biases. Heterogeneity of patient demographics. Risk of subjective bias (most studies do not use BNI to assess outcomes). Heterogeneity in GKS procedure.	Critically low
Texakalidis P et al. (2019) [14]	Low evidence level of the included studies. Heterogeneity in procedure details. Heterogeneity in follow-up. Lack of subgroup analyses in pain relief subject.	Low
Diana C et al. (2021) [7]	Treatment choice heterogeneity: based on patience preference and surgeon expertise. Lack of standardized outcome measure: risk of subjective bias. Heterogeneity in follow-up.	Low
Chen JN et al. (2021) [10]	Heterogeneity of the studies. Potential selection bias.	Low
Yan C et al. (2021) [11]	Potential selection and recall biases. Low evidence level of the included studies affects the pools effect estimates of the meta-analyses. Significant heterogeneity not fully explained by sensitivity and subgroup analyses. Lack of a study protocol (i.e., PROSPERO).	Critically low
Nascimento RFV et al. (2023) [13]	High heterogeneity in long-term pain relief, recurrence and complication assessment. Lack of a study protocol (i.e., PROSPERO). Low evidence level of the included studies.	Low

## Data Availability

Not applicable.

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
