# Peer review of "Outcome Comparison of Drug-Resistant Trigeminal Neuralgia Surgical Treatments—An Umbrella Review of Meta-Analyses and Systematic Reviews"

_brainsci, 2023, doi:10.3390/brainsci13040530_

Round 1

Reviewer 1 Report

The current manuscript is a quite interesting “review of reviews” regarding surgical treatments for drug-resistant trigeminal neuralgia. Although the collected evidence has its limitations, the authors made sure to thoroughly cover all the available options in this type of treatments. In my opinion, the manuscript should one small alteration: inserting, in the introduction section, a Figure representing the pathophysiology of trigeminal neuralgia (on a molecular level), and another Figure summarizing the included surgical treatments.

Author Response

Response to Reviewers

Reviewer 1

The current manuscript is a quite interesting “review of reviews” regarding surgical treatments for drug-resistant trigeminal neuralgia. Although the collected evidence has its limitations, the authors made sure to thoroughly cover all the available options in this type of treatments. In my opinion, the manuscript should one small alteration: inserting, in the introduction section, a Figure representing the pathophysiology of trigeminal neuralgia (on a molecular level), and another Figure summarizing the included surgical treatments.

Dear Reviewer thank you for your comments. As you suggested we inserted a Figure 1 (Summary of pathophysiological theories and molecular mechanisms involved in TN pathogenesis) and a Figure 2 (Summary of available surgical procedures for the treatment of drug-resistant TN)

Reviewer 2 Report

In this manuscript, the author conducted an umbrella review of meta-analyses and systematic reviews. However, the author can address the following comments to proceed further:

1. Why this review has been designed and what is the significance of this review?

2. How this review will be helpful for the scientific community? It is well known that MVD is an effective procedure for the treatment of intensely painful facial nerves. 

3. Why do GKS and SRS show the highest variation in terms of pain relief? Why radiosurgery shows the highest variation in terms of pain relief and a higher rate of delayed responses?

Author Response

Response to Reviewers

Reviewer 2

In this manuscript, the author conducted an umbrella review of meta-analyses and systematic reviews. However, the author can address the following comments to proceed further:

1. Why this review has been designed and what is the significance of this review?

Dear Reviewer thank you for your comment. As we stated in the introduction section the aim of this “umbrella review” was to provide an updated evaluation of the evidence reported in literature, comparing the results of the different surgical procedures for TN. In our opinion this types of studies are especially useful when different treatments (in our case surgical treatments) are available for a single pathology. 

2.How this review will be helpful for the scientific community? It is well known that MVD is an effective procedure for the treatment of intensely painful facial nerves.

Thank you for your comment. In our work we found that, although MVD appears the most effective treatment, the quality of evidence is homogenously scarce even in systematic reviews and meta-analyses (objects of this “umbrella review”). Thus, as we stated in the discussion and in the conclusion sections, to date we have no strong evidences in support of one surgical technique over another one. This study help the scientific community because the umbrella reviews are on the top of scientific evidence and the results of this umbrella review clearly stated the need of RCTs comparing the different surgical techniques for the TN treatment.

3. Why do GKS and SRS show the highest variation in terms of pain relief? Why radiosurgery shows the highest variation in terms of pain relief and a higher rate of delayed responses?

Thank you for your comment. As we stated in the discussion section GKS and SRS shows the highest variation in term of pain relief because there is wide variation among studies in regard of optimal radiation dose, location of delivery, number of isocenters and length of involved trigeminal nerve for radiation. The delayed effect of GKS and SRS (due to the type of radiobiological mechanisms of action of these techniques) is well known in the literature and reported by different studies.

Round 2

Reviewer 2 Report

Thank you for addressing the comments and highlighting the significance of the study. 

Now 2 Figures have been added that certainly enhance the information of the review. 

Now, I have 2 suggestions:

1. Please elaborate on the molecular mechanisms involved in TN pathogenesis presented in Figure 1. 

Author Response

Response to Reviewer 2

Thank you for addressing the comments and highlighting the significance of the study. Now 2 Figures have been added that certainly enhance the information of the review.

Now, I have 2 suggestions:

  1. Please elaborate on the molecular mechanisms involved in TN pathogenesis presented in Figure 1.

Thank you for your comment. We better explained the molecular mechanisms involved in TN pathogenesis adding the following paragraph in the introduction section: “From a molecular point of view different molecules have been advocated to be involved in TN pathogenesis. Recently we investigated the role of different inflammatory and neurodegenerative factors and found that neuron specific enolase (NSE) is involved in TN pathogenesis [1]. NSE is a glycolytic enzyme which is expressed in the cytosol of neurons and neuroendocrine cells. Elevated NSE can promote extra-cellular matrix degradation, inflammatory glial cell proliferation, and actin remodeling, thereby affecting migration of activated macrophages and microglia to the injury site and promoting neuronal cell death. Different types of insults may damage trigeminal nerve leading to Schwann’s cells degeneration and axonal dystrophy. According to our results, NSE could be upregulated in TN, thus transiting on the cell surface of neuronal cells where it contributes to trigger nerve damage and inflammation. By fact, we found elevated NSE in serum of TN affected patients [1]. Furthermore NSE concentration in CSF appeared strictly related to the severity of NVC, being highly expressed in case of nerve atrophy [3]”